# Efficacy and Safety of Quadrivalent Conjugate Meningococcal Vaccines: A Systematic Review and Meta-Analysis

**DOI:** 10.3390/vaccines11010178

**Published:** 2023-01-13

**Authors:** Andrea Conti, Gaia Broglia, Chiara Sacchi, Fabrizia Risi, Francesco Barone-Adesi, Massimiliano Panella

**Affiliations:** Department of Translational Medicine, Università del Piemonte Orientale, 28100 Novara, Italy

**Keywords:** meningococcal vaccine, quadrivalent meningococcal vaccine, invasive meningococcal disease, vaccine efficacy, vaccine safety, systematic review, meta-analysis

## Abstract

Over the last decades, different quadrivalent antimeningococcal vaccine formulations (diphteria toxoid conjugate, MenACWY-D; tetanus toxoid conjugate, MenACWY-TT; CRM_197_ protein conjugate, MenACWY-CRM) have been developed. However, their availability varies, both in terms of authorized formulations and of inclusion in vaccination schedules. Furthermore, several countries include only the monovalent meningococcal C (MenC) vaccine in their immunization programmes. Finally, there is currently no updated systematic review that directly compares the MenACWY formulations. Thus, we summarized the evidence on efficacy and safety through four parallel, independent systematic literature reviews with meta-analysis which included randomized controlled trials comparing the abovementioned vaccines. A total of 16 studies have been included. In terms of efficacy, MenACWY-TT outperformed MenACWY-D and MenACWY-CRM for A, W-135, and Y serogroups, while no significant difference was found for serogroup C. Furthermore, we did not find significant differences in efficacy between MenC and MenACWY-TT. Regarding the safety, we were able to perform a quantitative analysis only between MenACWY-TT and MenC, finding no significant differences. Similarly, among the different MenACWY formulations no relevant differences were identified. These findings suggest that MenACWY-TT could be preferable to other formulations to improve current vaccination programs and to better develop future immunization policies.

## 1. Introduction

Invasive meningococcal disease (IMD) is a severe condition caused by *Neisseria meningitidis*, characterized by a rapid onset and fatality rates up to 80% in untreated subjects [1]. Moreover, survivors often suffer from long term sequelae, such as hearing loss and amputations [2]. The poor clinical outcomes [1] and the relevant associated economic costs [3] characterize IMD as a major public health issue. In this context, vaccination is universally recognized as one of the most effective strategies to mitigate the incidence of IMD. Accordingly, vaccination campaigns have been strongly recommended [4,5], especially for the most susceptible subjects (namely, infants, children, and young adults) [6].

Several vaccines against *N. meningitidis* have been developed and commercialized over the last decades [7]. The introduction of a conjugate vaccine against serogroup A (*MenAfriVac*^®^, Meningitis Vaccine Project) in sub-Saharian Africa halved the number of suspected meningitis cases, proving the importance of massive immunization campaigns [8]. Similarly, vaccination programs with monovalent serogroup C meningococcal conjugate vaccines (MenC) have successfully reduced the burden of the disease in infants, older children, and adults in Europe [9]. Although these campaigns have significantly mitigated the IMD incidence, the broad use of MenC caused a relative increase of the cases associated with A, B, W-135, and Y serogroups [10]. In particular, the emergence of a hypervirulent meningococcal W-135 clone has pushed many developed countries to consider the administration of quadrivalent meningococcal vaccines against serogroups A, C, W-135, and Y (MenACWY) [10,11].

The story of MenACWY started in 2005, with the licensure of the diphtheria toxoid conjugate vaccine (MenACWY-D) in the United States, followed by the CRM_197_ protein conjugate vaccine (MenACWY-CRM) in 2010 and the approval of the tetanus toxoid conjugate vaccine (MenACWY-TT) in the European Union two years later [7]. However, despite these formulations have been available for several years, the vaccination strategies are very heterogeneous among the different developed countries. For example, MenACWY-TT is available in the market as two different products (*Nimenrix*^®^, Pfizer, and *MenQuadfi*^®^, Sanofi Pasteur), of which only *MenQuadfi*^®^ is available in the United States since 2020. Similarly, the MenACWY-D is not available in Europe [7]. In addition, only about 60% of European countries includes MenACWY in their national immunization plans and, among them, vaccination schedules are highly different in terms of inoculation ages and number of doses [12].

It is also noteworthy that, despite the long-standing story of MenACWY vaccines, there are still different gaps of knowledge in the relevant scientific literature. In particular, the last reviews comparing different types of MenACWY were conducted in 2014 [4,13]. While one mostly focused on polysaccharide meningococcal vaccines, which are not used anymore, and found only one study comparing MenACWY-TT and MenACWY-D [13], the other did not include MenACWY-CRM vaccines [4]. Furthermore, none of them performed any meta-analysis directly comparing the MenACWY vaccines investigated in our review. Thus, there is currently no updated and comprehensive systematic quantitative synthesis on these vaccines.

For these reasons, we systematically summarized the available evidence from randomized controlled trials on the efficacy and safety of the different formulations of MenACWY vaccines. We also carried out meta-analyses of the quantitative results.

## 2. Materials and Methods

### 2.1. Search Strategy

We conducted four parallel systematic reviews with meta-analysis of literature according to the Recommendations of the Preferred Reporting Items for Systematic Reviews and Meta-Analyses statement (PRISMA) [14]. The search was performed on the PubMed database on 30 October 2022. Table 1 shows the search strings and the objectives of each review.

Similar to several regulatory authorities, we considered the proportion of individuals presenting a serum bactericidal activity (SBA) title ≥1:8 using either human or rabbit complement assays one month after the vaccination as the main efficacy outcome [15]. SBA titers are an indirect measure of protection, and are considered the gold standard for infectious diseases with a low incidence rate such as IMD [5,16]. Therefore, we excluded studies considering different outcomes for efficacy (e.g., long-term antibody persistence). In addition, only papers describing clinical trials performed in healthy subjects and evaluating MenC, MenACWY-CRM, MenACWY-D, and MenACWY-TT vaccines were included. Apart from quadrivalent vaccines, we included MenC in our review as several European countries comprise only the latter in vaccination schedules [12] and a direct comparison can bolster the results of this review, especially in the light of the long standing history of MenC vaccination campaigns.

Studies evaluating quadrivalent meningococcal polysaccharide vaccine (MenPS) were excluded because MenPS is not used anymore in routine clinical practice. We also excluded studies performed after a booster dose since the majority of European countries recommend a single administration [12]. Finally, we excluded studies published in a language different from English and studies with the coadministration of meningococcal vaccine together with other vaccines.

### 2.2. Data Extraction, Analysis, and Synthesis

Extracted information was entered in a Microsoft Excel database; three researchers (C.S., G.B., F.R.) independently screened the articles to assess studies’ eligibility for inclusion. Inconsistencies were resolved after a discussion involving the whole research team. Gathered information was organized, and subsequently analyzed, according to the four abovementioned reviews (Table 2). In the first review, we compared the immunogenicity of MenACWY-CRM, MenACWY-DT, and MenACWY-TT. In the second one, we compared the immunogenicity of MenC to MenACWY. In the third one, we compared adverse effects between MenC and any other MenACWY. Finally, in the fourth one we compared adverse effects the different MenACWY vaccines. We conducted the quality appraisal of the included studies using the Cochrane Risk of Bias 2 Tool [17].

Where possible, we carried out a random-effect meta-analysis of the results of the different reviews. Results were reported as risk ratios of the outcome of interest (namely, SBA titer higher than the pre-defined threshold for immunogenicity, and the proportion of adverse events (AE) for safety). Regarding the safety comparison, the meta-analysis was stratified basing on the following subgroups: mild local reaction, severe local reaction, mild systemic reaction, and severe systemic reaction. This classification was based on the original, three-level grading of symptoms used among the majority of the studies included in Review 3. In detail, we considered as “mild” the reactions from grade 1 and 2, and “severe” the reactions reported as grade 3.

## 3. Results

PRISMA flowcharts of the screening process are available in Figure A1. Overall, 16 different studies were included (Table 2). All of them were randomized-controlled trials, of which ten were a phase 3 and six phase 2 trials. Four studies were double-blind, three single-blind and nine used on open label design. The number of enrolled participants per trial varied from 202 to 3344, with a median of 1000 patients. All studies evaluated the vaccine efficacy, while 15 of them also evaluated safety. The most represented countries were the United States (8 studies), followed by Germany (4), Austria, Canada, Finland (3), and France, Greece, Italy, Puerto Rico, Latin America, Denmark (1).

With regards to the quality appraisal, the included studies showed a low to moderate risk of bias. Indeed, some of the studies used an open label design, in which participants are aware of the type of vaccine administered, resulting in a moderate risk of bias for four studies. Only one study [33] showed moderate risk of bias due to a non-completely clear data analysis plan. The complete risk of bias assessment matrix is available in Figure A2. The results of the four reviews are reported in the following paragraphs.

### 3.1. Review 1: Efficacy Comparison among Different MenACWY Vaccines

A total of ten studies compared the efficacy of different quadrivalent vaccines among them. Meta-analytic pairwise comparisons of the different serogroups are reported here.

#### 3.1.1. MenACWY-TT vs. MenACWY-CRM

As shown in Figure 1, three studies compared the efficacy of MenACWY-TT against MenACWY-CRM [18,20,21]. Overall, MenACWY-TT showed a higher efficacy than MenACWY-CRM when all the serogroups were considered (RR: 1.12, 95% CI 1.05–1.19). A statistically significant effect was found for serogroup A (RR: 1.09, 95% CI: 1.04–1.15), W (RR: 1.09, 95% CI: 1.07–1.12), and Y (RR: 1.09, 95% CI: 1.06–1.11), but not for C (RR: 1.23, 95% CI: 0.99–1.54). Notably, the heterogeneity of the estimates for serogroups C was also particularly high (I^2^: 96%).

#### 3.1.2. MenACWY-TT vs. MenACWY-D

Three studies compared the efficacy of MenACWY-TT against MenACWY-D [19,22,24]. The metanalysis (Figure 2) was performed only on two studies [19,22], as Halperin et al. [24], presented the proportion of subjects with SBA titers ≥ 1:8 only using a graphical chart, from which was not possible to gather original data. Also in this case, a statistically significant effect was found for serogroup A (RR: 1.09, 95% CI: 1.02–1.17), W (RR: 1.14, 95% CI: 1.05–1.24), and Y (RR: 1.13, 95% CI: 1.10–1.16), but not for C (RR: 1.10, 95% CI: 0.86–1.42). Moreover, the heterogeneity for serogroup C was very high (I^2^: 99%).

#### 3.1.3. MenACWY-D vs. MenACWY-CRM

The efficacy of MenACWY-D vs. MenACWY-CRM was investigated in four studies [23,25,30,31]. MenACWY-D was significantly less effective than MenACWY-CRM for serogroups W (RR: 0.89, 95% CI: 0.81–0.98) and Y (RR: 0.78, 95% CI: 0.67–0.90), while no significant difference was found for serogroups A (RR: 1.00, 95% CI: 0.91–1.10) and C (RR: 0.96, 95% CI: 0.90–1.01) (Figure 3). For all the serogroups, a substantial heterogeneity in the estimates was found.

### 3.2. Review 2: Efficacy Comparison of Quadrivalent MenACWY vs. MenC Vaccines

Limitedly to serogroup C, six studies [26,27,28,29,32,33] compared the efficacy of MenACWY-TT against MenC. The RR was 1.00 (95% CI 1.00–1.01) with very low heterogeneity (Figure 4).

We did not find any study comparing MenC with MenACWY-CRM, or with MenACWY-D.

### 3.3. Review 3: Safety Comparison of MenACWY vs. MenC

Five studies compared the safety of MenACWY-TT against MenC [26,28,29,32,33]. Similar to Review 2, none of the included studies compared the safety of MenACWY-D or MenACWY-CRM versus MenC. As shown in Figure 5, we did not find significant differences in the frequency of AE (RR: 1.02, 95% CI: 0.90–1.15). Also, subgroup analysis did not show substantial differences (Local mild reactions, RR: 1.12, 95% CI: 0.93–1.35; Local severe reactions, RR: 1.10, 95% CI: 0.30–4.06; Systemic mild reactions, RR: 0.97, 95% CI: 0.81–1.15; Systemic severe reactions, RR: 0.87, 95% CI: 0.23–3.25) Heterogeneity ranged between 0% and 53% in the different analyses. One of the studies was not included in the meta-analysis as it was based on a different AE classification [29].

### 3.4. Review 4: Safety Comparison among Different Quadrivalent Meningococcal Vaccines

Ten studies matched the criteria for the fourth review [18,19,20,21,22,23,24,25,30,31]. Due to the different reported outcomes and observation periods which were considered, it was not possible to perform a meta-analysis. Therefore, we summarized results in a descriptive way.

Three studies compared MenACWY-TT and MenACWY-CRM safety profiles [18,20,21]. Baccarini et al. [18] monitored solicited AE, systemic reactions, and serious AE up to 30 min, 7 days, and 30 days, respectively. Chang et al. [21] monitored immediate reactions (within 30 min after inoculation) and delayed reactions (within 180 days after the inoculation), while Bona et al. [20] monitored solicited AE for 7 days, unsolicited AE for 29 days and medically attended AE for the study period. All the studies found a comparable safety profile for MenACWY-TT and MenACWY-CRM.

The safety of MenACWY-D compared to MenACWY-CRM was assessed by four studies [23,25,30,31]. All the studies found a comparable safety profile for AE within 7 days after the inoculation. Regarding immediate reactions, Stamboulian et al. [31] found in the 56–65 age group a higher number of reports for unsolicited AE in MenACWY-CRM than MenACWY-D. Halperin et al. [23] reported, in the 6–10 age group, less fever and more erythema reactions for the MenACWY-CRM than the MenACWY-D.

Three studies compared the safety of MenACWY-D to MenACWY-TT [19,22,24], and none of them reported relevant differences in safety profiles. While Halperin et al. [24] recorded solicited AE for three days, the other two studies [19,22] conducted a 7-days monitoring.

## 4. Discussion

In the present study, we used data from 16 RCTs including more than 20,000 individuals to investigate the efficacy and safety of quadrivalent meningococcal vaccines. As a major result, we found that overall MenACWY-TT was significantly more effective than MenACWY-CRM and MenACWY-D. In particular, MenACWY-TT outperformed the other vaccines on A, W-135, and Y serogroups, while for the serogroup C no significant difference was found. The comparison between MenACWY-D and MenACWY-CRM pointed out the overall superiority of the latter, while subgroup analyses identified a significant greater efficacy for serogroups W-135 and Y. Furthermore, our results also found a similar efficacy of MenACWY-TT and MenC regarding the protection against serogroup C.

Interestingly, immunization policies of most developed countries currently do not favor a specific type of quadrivalent vaccine. For example, in the United States, the Advisory Committee on Immunization published the latest meningococcal vaccination guidelines two years ago, recommending quadrivalent vaccination but not suggesting a specific MenACWY formulation [34]. In the same way, MenACWY-TT and MenACWY-CRM are equally recommended among European countries [12], even though both have been available for more than 10 years [7] during which several studies compared the two formulations.

As a second point, vaccination strategies must take in consideration the safety and the occurrence of AEs. Our review also highlights that all MenACWY vaccines have a reassuring safety profile, with little or no differences between the different types. However, it should be noted that we were able to quantitatively compare adverse reactions only among MenACWY-TT and MenC due to differences in observation periods and outcome in the studies evaluating the other types of vaccines. In this respect, the use of a more standardized classification of adverse reactions in future studies would be very useful to thoroughly evaluate the safety profiles. Furthermore, our review considered the AEs occurred only within 30 days after the inoculation. Despite this is considered the standard timespan for monitoring AEs [35], observational studies based on large samples and considering a wider period of time could help to investigate the potential onset of infrequent or long-term adverse effects.

Another factor that should be taken in designing and implementing vaccination programmes is the cost-effectiveness ratio. Although MenACWY is one of the most expensive vaccine [36], some studies suggest that its implementation among developed countries could be cost-effective [37,38]. For example, it has been estimated that vaccinating the 15–19 years old Australians with MenACWY could lead to 2058 quality adjusted life years (QALY) gained and 114 million Australian dollars of direct and indirect costs saved [39]. Similar results were obtained in Canada, where the introduction of MenACWY among adolescents could save 4291 QALY and 46 million Canadian dollars [40]. Despite our review does not aim to assess the economic aspects of MenACWY vaccination, it is interesting to observe that the MenACWY-TT price is only 2% higher than the MenACWY-CRM, at least in the United States [41]. Under this view and considering the superiority of MenACWY-TT observed in our study, we think that more studies comparing the cost-effectiveness of the different types of MenACWY are warranted.

It is worth mentioning that our review presents some limitations. First, the number of included studies is small. Indeed, despite the large number of participants, analyses were performed on subgroups of 2 to 6 original studies. Thus, meta-analytic estimates are heavily dependent from the results of specific studies. For example, the possible presence of confounding in one or more studies could have affected the results. However, as we included only randomized controlled trials, we are reasonably confident that no significant differences were present among the two arms of each study. In addition, despite the antibody development can vary with subject age, we were not able to broadly perform age-based subgroup analyses due to heterogeneity among studies. Moreover, some of the studies were conducted following an open label design. Although it is unlikely that adverse reaction reporting was affected by the absence of participants’ blinding, we cannot completely exclude this phenomenon. In terms of efficacy, it should be noted that studies comparing MenC against MenACWY-TT used rabbit complement SBA, while the others considered human complement SBA, which has a different titer wane profile [42]. However, in each meta-analysis all the studies used the same type of SBA as a proxy measure of protection. Thus, we do not expect that the results have been affected by it.

Furthermore, our review does not include studies on vaccines against meningococcal serogroup B, that causes the majority of IMD cases in developed countries [12]. Indeed, currently only monovalent vaccines against this serogroup (*Bexsero*^®^, GSK, and *Trumenba*^®^, Pfizer) are licensed, not allowing a direct comparison to MenACWY. However, it is worth mentioning that a pentavalent MenABCWY vaccine is currently under development, which is expected to simplify and improve vaccinations programmes wordwide [43].

As final consideration, the abovementioned emergence of certain serogroups after the introduction of MenC, as well as the changing migration flows that could influence the current serogroups prevalence [44], highlight the importance of a comprehensive approach in controlling all the serogroups, together with the implementation of evidence-based vaccination strategies. In this regard, the adoption of quadrivalent formulations could substantially help to mitigate the incidence of IMD.

## 5. Conclusions

Among the MenACWY vaccines, the MenACWY-TT proved to be more effective than MenACWY-D and MenACWY-CRM, and showed a comparable effectiveness to MenC for serogroup C. Moreover, the safety profiles are similar among all the investigated vaccines. These results, together with the changing epidemiological landscape of meningitis, suggest that the adoption of MenACWY-TT instead of other formulations could be taken in consideration for future immunization policies.

## Figures and Tables

**Figure 1 vaccines-11-00178-f001:**
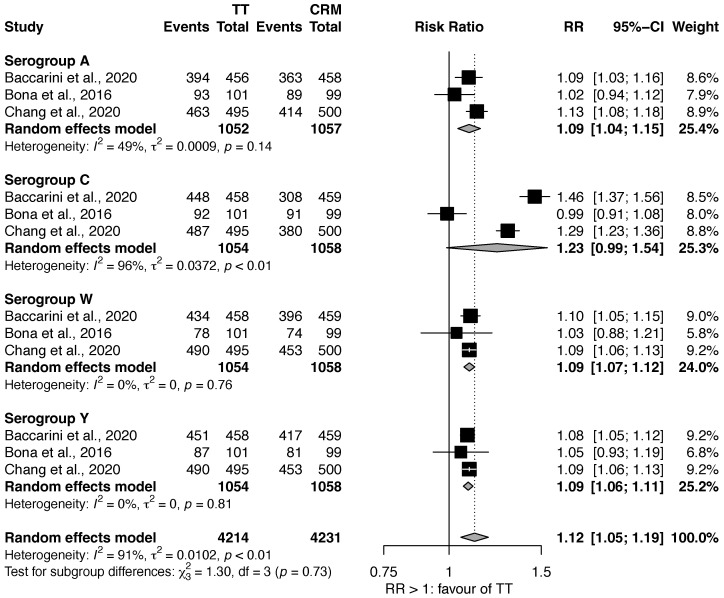
Efficacy of MenACWY-TT vs. MenACWY-CRM [18,20,21].

**Figure 2 vaccines-11-00178-f002:**
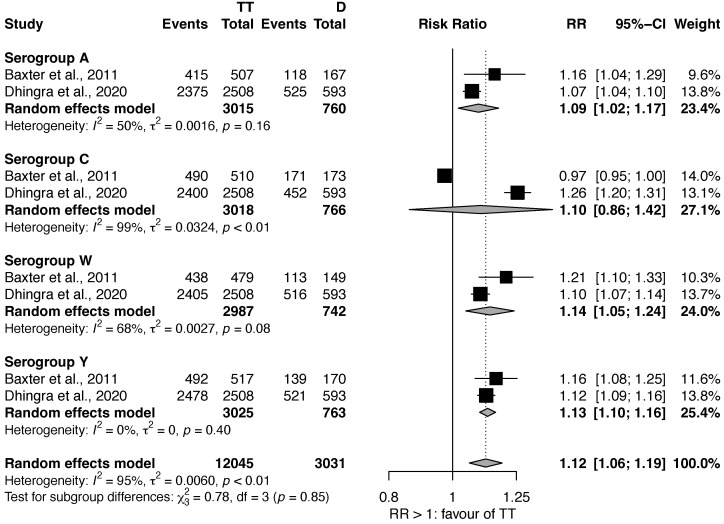
Efficacy of MenACWY-TT vs. MenACWY-D [19,22].

**Figure 3 vaccines-11-00178-f003:**
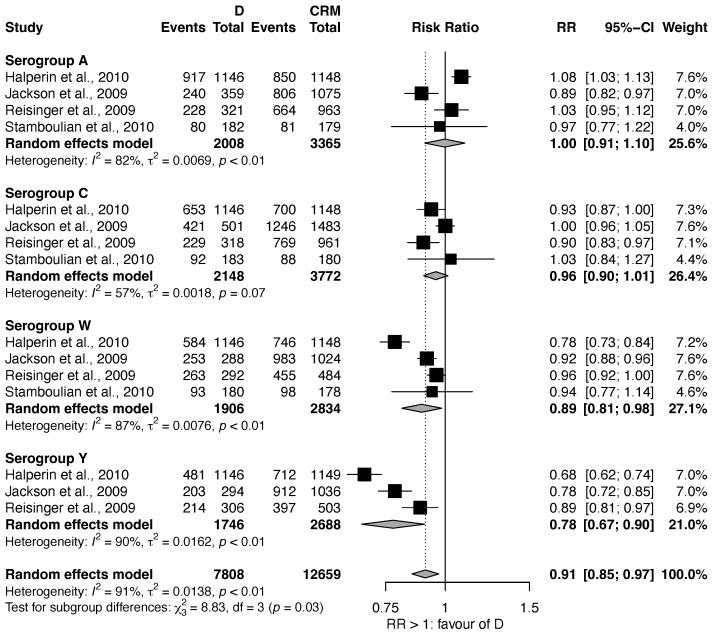
Efficacy of MenACWY-D vs. MenACWY-CRM [23,25,30,31].

**Figure 4 vaccines-11-00178-f004:**
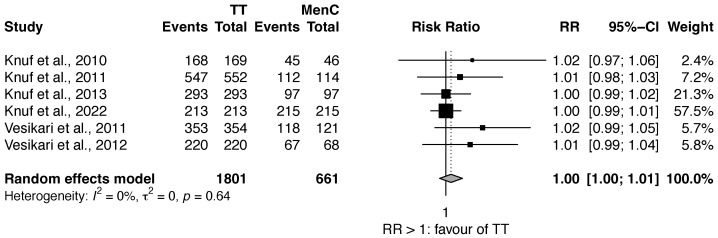
Efficacy of MenACWY-TT vs. MenC (Serogroup C) [26,27,28,29,32,33].

**Figure 5 vaccines-11-00178-f005:**
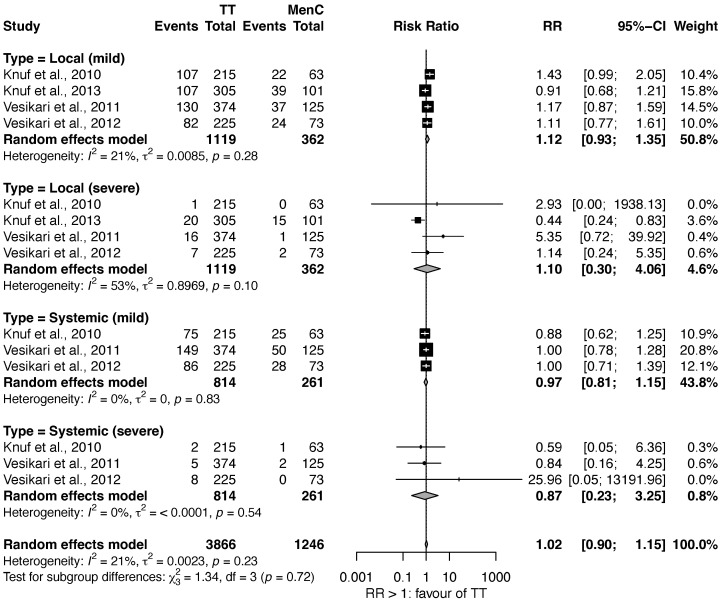
Safety of MenC vs. MenACWY-TT [26,28,32,33].

**Table 1 vaccines-11-00178-t001:** Search strings and objectives of the different reviews.

Review	String	Objective
1	“Meningococcal Vaccines”[Mesh] AND quadrivalent	To compare the efficacy of different quadrivalent vaccines.
2	“Vaccines”[Majr] AND menc	To compare the efficacy of quadrivalent and monovalent vaccines.
3	“Meningococcal Vaccines/adverse effects”[Mesh] OR (“Meningococcal Vaccines”[Mesh] AND safety)	To compare safety of monovalent and quadrivalent vaccines.
4	“Meningococcal Vaccines/adverse effects”[Mesh] OR (“Meningococcal Vaccines”[Mesh] AND safety)	To compare safety among different quadrivalent vaccines.

**Table 2 vaccines-11-00178-t002:** Studies included in the systematic reviews. N: number of enrolled subjects; E: efficacy; S: safety; Review: number of the review(s) that includes the study (see Table 1); Ph2: phase 2 randomized controlled trial; Ph3: phase 3 randomized controlled trial.

Study	Countries	N	Subject Age	Aim	Review	Vaccines	Study Design
Baccarini et al. [18]	United States, Puerto Rico	1000	2–9 years	E + S	1, 4	MenACWY-TT, MenACWY-CRM	Ph3, double-blind
Baxter et al. [19]	United States	784	10–25 years	E + S	1, 4	MenACWY-TT MenACWY-D,	Ph2 single-blind
Bona et al. [20]	Italy	202	12–15 months	E + S	1, 4	MenACWY-TT, MenACWY-CRM	Ph2 single-blind
Chang et al. [21]	United States	1715	10–17 years	E + S	1, 4	MenACWY-TT, MenACWY-CRM	Ph2 open-label
Dhingra et al. [22]	United States	3344	10–55 years	E + S	1, 4	MenACWY-TT MenACWY-D,	Ph3 modified double-blind
Halperin et al. [23]	United States, Canada	2907	2–5 years	E + S	1, 4	MenACWY-D, MenACWY-CRM	Ph3 single-blind
Halperin et al. [24]	United States, Canada	1016	10–25 years	E + S	1, 4	MenACWY-TT, MenACWY-D	Ph2 observer-blind
Jackson et al. [25]	United States	2180	11–18 years	E + S	1, 4	MenACWY-D MenACWY-CRM,	Ph3 observer-blind
Knuf et al. [26]	Germany, Austria	508	1–5 years	E + S	2, 3	MenACWY-TT, MenC	Ph2 double-blind
Knuf et al. [27]	Austria, Germany, Greece	793	12–23 months	E + S	2	MenACWY-TT, MenC	Ph3 open
Knuf et al. [28]	Germany, France	413	2–10 years	E	2, 3	MenACWY-TT, MenC	Ph3 open
Knuf et al. [29]	Denmark, Germany, Finland	707	12–23 months	E + S	2, 3	MenACWY-TT, MenC	Ph3 double-blind
Reisinger et al. [30]	United States	1359	19–55 years	E + S	1, 4	MenACWY-D, MenACWY-CRM	Ph3 open
Stamboulian et al. [31]	Latin America	2505	19–65 years	E + S	1, 4	MenACWY-D, MenACWY-CRM	Ph3 observer-blind
Vesikari et al. [32]	Finland	1000	12–23 months	E + S	2, 3	MenACWY-TT, MenC	Ph3 single-blind
Vesikari et al. [33]	Finland	304	12–23 months	E + S	2, 3	MenACWY-TT, MenC	Ph2 open

## Data Availability

Data is contained within the article.

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
