# Peer review of "Efficacy and Safety of Quadrivalent Conjugate Meningococcal Vaccines: A Systematic Review and Meta-Analysis"

_vaccines, 2023, doi:10.3390/vaccines11010178_

Round 1

Reviewer 1 Report

The manuscript of Conti et al. "Efficacy and Safety of Quadrivalent Conjugate Meningococcal Vaccines: a Systematic Review and Meta-Analysis" evaluates the efficacy and safety of quadrivalent antimeningococcal vaccine formulations (MenACWY-TT, MenACWY-CRM, and MenACWY-D) and a monovalent one (MenC) by independent systematic literature reviews with meta-analysis, which included randomized controlled trails, comparing the vaccines formulations. After the analyses and based on criteria of inclusion, 16 studies were considered. In terms of efficacy, MenACWY-TT outperformed the other quadrivalent vaccines for A, W-135, and Y serogroups, while no significant difference was found for serogroup C. There was also no significant difference between MenACWY-TT and MenC.  Regarding the safety, no significant differences were observed among the formulations. The conclusion of the study indicated the superior efficacy of MenACWY-TT compared to the other formulations and the main advantage of the quadrivalent formulations when compared to the MenC. The study is well done, and a valuable contribution to clarify the efficacy of the main meningococcal vaccine formulations available. One of the limitations of the study was recognized by the authors and is related to the low number of studies (16) included in the analyses. However, the is due to the low number of studies performed so far comparing the quadrivalent formulations and the date described support the main conclusions of the manuscript. Therefore, as mentioned, this study contributed with important information regarding the efficacy and safety of meningococcal vaccine formulations.

Author Response

We would like to thank the Reviewer for the time spent in assessing our manuscript and for the nice comment. In this new version of the manuscript, we have addressed the issues pointed out by the other Reviewers. Changes are highlighted in yellow.

Reviewer 2 Report

Thank you for your submission. The manuscript is interesting and well-written. Was the protocol registered with Prospero?

Should section 3.2/figure 4 be clearer that this is presumably looking only at serogroup C?

It should be acknowledged as a limitation that a proxy measure (one-month SBA titer) was used, rather than actual meningococcal infections, as the measure of efficacy; the observed differences in titers may not translate into differences in likelihood of clinical infection.

Author Response

We would like to thank the Reviewer for the time spent in assessing our manuscript. Please find below our responses to the comments. Changes in the manuscript are highlighted in yellow.

Thank you for your submission. The manuscript is interesting and well-written. Was the protocol registered with Prospero?
The protocol was not registered in Prospero.

Should section 3.2/figure 4 be clearer that this is presumably looking only at serogroup C?
We have rephrased the text in section 3.2 and Figure 4 caption to clarify this and to avoid misunderstandings.

It should be acknowledged as a limitation that a proxy measure (one-month SBA titer) was used, rather than actual meningococcal infections, as the measure of efficacy; the observed differences in titers may not translate into differences in likelihood of clinical infection.
We have rephrased this concept both in “materials and methods” and in “discussion” sections.

Reviewer 3 Report

Review by et al covers systemic review and meta- analysis of efficacy and safety of Quadrivalent Conjugate Meningococcal Vaccines. Review covers comparative analysis of 6 MenACWY formulations by analyzing sixteen studies covering 20000 individuals to investigate the efficacy and safety of quadrivalent meningococcal vaccines. The need of the study is clearly justified.  The review uses a licensure criteria of SBA titers which is also very relevant for analysis. The analysis covered four parallel systematic reviews with meta-analysis of literature according to the Recommendations of the Preferred Reporting Items for Systematic Reviews and Meta-Analyses.  The most represented countries were the United States (8 studies), followed by Germany (4), Austria, Canada, Finland (3), and France, Greece, Italy, Puerto Rico, Latin America, Denmark (1). The study was carried out under following objectives:

a)       To compare the efficacy of different quadrivalent vaccines.

-          In this review- immunogenicity of MenACWY-CRM, MenACWY-DT, and MenACWY-TT formulations were compared.

b)      To compare the efficacy of quadrivalent and monovalent vaccines.

- compared the immunogenicity of MenC to MenACWY

c)       To compare safety of monovalent and quadrivalent vaccines.

- adverse effects between MenC and any other MenACWY

d)      To compare safety among different quadrivalent vaccines

- adverse effects the different MenACWY vaccines

The study suggests following conclusions under four objectives:

-          MenACWY-TT outperformed MenACWY-D and MenACWY-CRM for A, W-135, 10 and Y serogroups, while no significant difference was found for serogroup C.

-          For review b-no significant difference between efficacy of Men C and Men ACYW-TT.

-          For review c-no significant difference between efficacy of Men C and Men ACYW-TT.

-          For review-d- no significant differences among quadrivalent vaccines.

Comments

The study subject is relevant. The study chose quadrivalent formulations which are also relevant to current scenario. The study covers a good number of reported studies. The introduction, materials section is clear.

However, study has following major gaps and limitations and thus is not suitable for publication in its current form:

-          The analysis appears very superficial. A serious attempt to discuss the findings is missing. Discussion section looks very general and does not discuss the findings in details.

-          The rationale for selection of objectives is not clear. For instance, why the study chose to compare Men C with Men A, C, W and Y in terms of safety and efficacy is not clear.

-          Why only SBA titers were chosen. Why the IgG titers were not considered for analysis.

-          The study suggests that no there is no meta-analysis of conjugate vaccines in prior art. We would like to highlight the following publication on such prior art-

Pellegrino et al. Immunogenicity of meningococcal quadrivalent (serogroup A, C, W135 and Y) tetanus toxoid conjugate vaccine: systematic review and meta-analysis. Pharmacological Research, 30 Oct 2014, 92:31-39.

Additionally, the study completely missed the monovalent Meningococcal Serogroup A conjugate vaccines in the study. Serogroup A is the most prevalent serotype in sub-Saharan Africa and is responsible for epidemics in sub-Saharan Africa. The introduction of serotype A conjugate vaccines in Africa have completely eradicated the disease. It appears from the study that the authors have chosen studies relevant or representative of formulations which are licensed in developed world. It is to be noted that Meningitis caused by Meningococcal group is mostly prevalent in sub-Saharan Africa. The review should have considered the vaccine efficacy in the African populations where the disease is prevalent.

The study excluded many parameters-no attempts to collate data from different parameters like age, concomitant vaccines. The authors could have done an exhaustive effort to analyze the data. Such attempt is missing.

Author Response

We would like to thank the Reviewer for the time spent in assessing our manuscript. Please find below our responses to the comments. Changes in the manuscript are highlighted in yellow.

The analysis appears very superficial. A serious attempt to discuss the findings is missing. Discussion section looks very general and does not discuss the findings in details.
Despite the Reviewer did not provide specific suggestions on which part(s) of the discussion needs improvements, we revised the discussion according also to the comments of the other reviewers.

The rationale for selection of objectives is not clear. For instance, why the study chose to compare Men C with Men A, C, W and Y in terms of safety and efficacy is not clear.
We decided to include also MenC as several European countries include only this vaccine (eventually together with MenB) in vaccination schedules (reference #11 of the paper). We believe that the comparison of the MenACWY to the current vaccination strategy (based on MenC) implemented in different European countries could push the latters to consider the introduction of the quadrivalent MenACWY, in the light of similar safety profile. We modified the Methods section to better clarify why we included MenC.

Why only SBA titers were chosen. Why the IgG titers were not considered for analysis.
We decided to consider only SBA titers as they are considered the gold standard for assessing the vaccine-induced protection (reference #5 of the manuscript) and are also predictive of human protection [1]. Furthermore, SBA is considered for vaccine licensure in the US [2] and in Europe [3]. We have better clarified this in the introduction.

  1. Goldschneider I, Gotschlich EC, Artenstein MS. Human immunity to the meningococcus. I. The role of humoral antibodies. J Exp Med. (1969)
  2. ACIP Guidelines
  3. EMA - MenQuadfi Summary of Product Characteristics

The study suggests that no there is no meta-analysis of conjugate vaccines in prior art. We would like to highlight the following publication on such prior art: Pellegrino et al. Immunogenicity of meningococcal quadrivalent (serogroup A, C, W135 and Y) tetanus toxoid conjugate vaccine: systematic review and meta-analysis. Pharmacological Research, 30 Oct 2014, 92:31-39.
We thank the reviewer for the suggestion. We added this piece of information in the introduction.

Additionally, the study completely missed the monovalent Meningococcal Serogroup A conjugate vaccines in the study. Serogroup A is the most prevalent serotype in sub-Saharan Africa and is responsible for epidemics in sub-Saharan Africa. The introduction of serotype A conjugate vaccines in Africa have completely eradicated the disease. It appears from the study that the authors have chosen studies relevant or representative of formulations which are licensed in developed world. It is to be noted that Meningitis caused by Meningococcal group is mostly prevalent in sub-Saharan Africa. The review should have considered the vaccine efficacy in the African populations where the disease is prevalent.
Our study purposely focused on the formulation used in the developed world, and specifically on MenACWY vaccine, as its main aim was to assess the quadrivalent vaccine efficacy and safety. We are aware about the relevance of serogroup A in the African population and, considering the pivotal role of MenA vaccines, we preferred to not mix the two different topics. Indeed, we believe that a similar research approach could be applied to the MenA formulations used in African countries, but it deserves a dedicated manuscript, rather than including it in our review.

The study excluded many parameters-no attempts to collate data from different parameters like age, concomitant vaccines. The authors could have done an exhaustive effort to analyze the data. Such attempt is missing.
In the inclusion criteria, we explicitly excluded studies in which the investigated vaccines were co-administered with others. Regarding age, unfortunately studies were extremely heterogeneous for this parameter (please see Table 2 of the manuscript). In detail, an age-based subgroup analysis was possible only for the studies comparing MenACWY-TT with MenC in the 12-23 months age group (4 studies). We added these analyses in the manuscript (figure A3-A4 of the Appendix A). No other studies could be grouped considering subject’s age. Moreover, we recognize the importance of having original studies comparable between each other. Therefore, we pointed out this limitation in the discussion.

Round 2

Reviewer 3 Report

The revised manuscript submitted by the authors were reviewed. The responses against the observations were also reviewed. It was noted that the major comment that the authors made were that the manuscript is restricted to formulations licensed in Europe. I noted this point. However, I am still not convinced about the value addition of this manuscript to the scientific community as we all are aware from prior art that tetanus conjugates are more immunogenic. What is the other value addition that this analysis is making is still not clear from the discussion or from author responses. 

Author Response

Dear Reviewer, we wish to clarify the relevance of our study. Despite other published studies already suggested the superiority of MenACWY-TT, we believe that there were gaps in the literature synthesis. Indeed, the latest similar but slightly different reviews were published eight years ago (references 4 and 12 of the manuscript). Please note that our review includes five studies (references 18, 20, 21, 22, 29 of the manuscript) published after 2014, subsequently not included in the previous literature syntheses. Furthermore, these two reviews do not contain any meta-analysis directly comparing all the quadrivalent vaccines included in our study: Zahlanie et al. did not perform any quantitative synthesis, while Pellegrino et al. were not able to perform a meta-analysis on MenACWY-TT vs MenACWY-D as they retrieved only one study. Furthermore, they did not compare MenACWY-DT with MenACWY-CRM. We synthesized these concepts in the last revision of the manuscript, lines 54-59 (in yellow), following your suggestion to improve the text given during review round 1.